# The Effects of a Macronutrient-Based Diet and Time-Restricted Feeding (16:8) on Body Composition in Physically Active Individuals—A 14-Week Randomised Controlled Trial

**DOI:** 10.3390/nu13093122

**Published:** 2021-09-06

**Authors:** Eduard Isenmann, Joshua Dissemond, Stephan Geisler

**Affiliations:** 1Department of Fitness and Health, IST-University of Applied Sciences, 40233 Dusseldorf, Germany; dissemond.j@web.de (J.D.); sgeisler@ist-hochschule.de (S.G.); 2Department of Molecular and Cellular Sports Medicine, Institute for Cardiovascular Research and Sports Medicine, German Sports University, 50933 Cologne, Germany

**Keywords:** time-restricted feeding, continuous energy restriction, macronutrient-based diet, intermitted fasting, body composition, lean body mass, fat mass

## Abstract

The number of people suffering from being overweight or obese has risen steadily in recent years. Consequently, new forms of nutrition and diets were developed as potential solutions. In the last years, the time-restricted feeding and continuous energy restriction via macronutrient-based diets were increasingly popular. Both diets were exclusively studied separately. A comparison of the two diets for people with a high body mass index despite regular physical activity has not yet been studied in detail. Therefore, this study aimed to compare the effects of these two diets on body composition and adherence. For this study, a total of 42 subjects (m = 21, f = 21) with a BMI above 25 were recruited from a local fitness gym. After a two-week familiarisation period, one of the two diets was followed over 14 weeks. Dietary behaviour was monitored throughout the period with a food diary. The primary measurement parameters were body weight, lean body mass, fat mass, body mass index, and waist and hip circumference. In addition, adherence was assessed and calculated by food diary and questionnaire. In total, the data of 35 participants (m = 14, f = 21) were analysed. Significant reductions in body weight, fat mass, body mass index, and waist and hip circumference were observed in both groups (*p* < 0.05). No significant change could be observed in lean body mass in either category. No group and gender differences were detected in any of the primary parameters. For the secondary parameters, a significantly higher adherence was observed in the time-restricted feeding group (*p* < 0.05). In addition, it can be assumed that an adherence of 60–70% cannot lead to positive changes in body composition. In conclusion, there were no differences between the two diets on the primary parameters. However, it seemed that time-restricted feeding can be better implemented in everyday life, and an adherence of more than 70% is required for both diets to prove effective.

## 1. Introduction

The number of people suffering from a body mass index within the range of “overweight” and “obese” has steadily increased in the last several decades. The western world in particular is struggling with this problem. Research by the Organisation for Economic Co-operation and Development (OECD) has shown that every second citizen of member states is overweight, and every sixth citizen is obese [1]. Consequently, the demand for effective diets has also increased. A variety of these has been reviewed in studies [2,3,4,5,6]. Two dietary strategies have become increasingly popular in recent years: the time-restricted feeding diet (TRF) and the macronutrient-based diet (MBD) with continuous energy restriction (CER).

MBD is a special form of CER. MBD differs from other CER diet methods in that the macronutrients fat, protein, and carbohydrates are set at a fixed gram value and the user tracks their diet with a food diary. The nutrient distribution is based on the recommendations of the International Society of Sports Nutrition for low-fat diets: carbohydrates 45–65%, fat 20–35%, proteins 10–35% [7]. The food source plays only a subordinate role. In addition, a specific calorie deficit of approximately 300–500 kcal is defined and should not be exceeded [8]. However, the effects of the dietary strategy have not yet been studied in detail. Initial studies only examined the effects of MBD and general CER on total calorie intake and macronutrient distribution [9].

In contrast, the evidence on the effect of TRF has been more widely examined. TRF is a special form of intermittent energy restriction, which differs from the other intermittent fasting methods in that energy intake is limited to a time window of 4–10 h. There is no fasting day, but rather a specific time window within which fasting takes place [10]. The best-known method of TRF is the 16:8 method [10]. In the 16:8 method, no food is consumed for 16 h, and the time window for eating is set for 8 h. A specific number of meals or calories is not prescribed. The TRF method has already been investigated in a few studies concerning the change of the primary measures body weight (BW), fat mass (FM), body fat percentage (BF%), body mass index (BMI), lean body mass (LBM), waist circumference (WC), and hip circumference (HC) [11,12,13,14,15,16,17,18,19,20,21]. The majority of studies examined the effects in mainly sick, overweight, or obese persons [11,12,13,14,15,16,17]. On average, the participants reduced their BW by −2.6% to −4.0%, FM by −4.0% to −5.9%, BMI by −1.09 kg/m^2^ to −1.15 kg/m^2^, and WC by −4.46 cm over a period of 8–16 weeks [11,12,13,14,15,16,17]. Moreover, a daily caloric deficit of 200−550 kcal was observed by using the different TRF methods [11,12,13,14,15,16,17]. Moreover, only a few studies investigated the TRF method in healthy, normal-weight individuals in terms of body composition and combination with resistance training [18,19,20,21]. Similar effects in overweight participants were observed for the parameters BW, FM, and LBM [19,20,21].

However, initial studies have only compared CER, but not MBD, with the TRF strategy. Both diets were tested for their effects on BW, FM, and LBM during a 4-week strength training program with a caloric deficit of 25% for both groups based on the subjects’ total daily energy requirements [18]. All participants completed a whole-body strength training three times a week [18]. Both groups had significant changes in BW (TRF −0.5%; CER −1.6%), BF% (TRF −1.6%; CER −1.5%), and FM (TRF −8.9%; CER −138.5%) [18]. However, no group differences were detected. Studies comparing TRF with MBD strategy are currently not available. In addition, adherence and participant questionnaires have hardly been considered in previous studies.

Therefore, the aim of this study was to investigate the effects of TRF and MBD strategy on body composition and adherence to both dietary strategies in healthy, physically active participants. For this purpose, a randomised controlled intervention study was conducted over 14 weeks.

## 2. Methods

### 2.1. Experimental Procedure

The study design was divided into three phases: a 2-week familiarisation phase (phase 1), an 8-week intervention phase (phase 2), and a 6-week independent phase without specific advice (phase 3). Before the 2-week familiarisation phase, the subjects were informed by a nutrition coach about the basics of nutrition (macronutrients, micronutrients, energy intake) and the two different diet strategies. In addition, the participants received instructions regarding the measurements and the handling of the food database (FDDB) Extender food diary (Food Database GmbH, 28217 Bremen, Deutschland), as well as the inclusion criteria of the study (T-1). At the beginning of the familiarisation phase (T0), all anthropometrical parameters were detected. Familiarisation was used exclusively to obtain a routine for dietary documentation and to reduce the influence of documentation on dietary behaviour. After two weeks, the measurements were repeated (T1), and the subjects were assigned to one of the two diets. To distribute the groups equally, stratified randomisation by gender, BMI, and physical activity was used. After T1, the 8-week dietary change and documentation were carried out. All participants recorded their daily food intake using the diaries and sent these records to the nutrition coach at the end of each week. In case of individual questions, they could also contact the nutrition coach. After phase 2, anthropometric parameters were measured (T2), followed by a six-week independent phase without specific instructions and communication with the nutrition coach. The participants were encouraged to continue and document their dietary patterns as best as possible. Finally, the anthropometric parameters were measured again (T3), the documentation of the diet in the last phase was reviewed, and a questionnaire was administered regarding their experience with the diet. The entire study design can be seen in Figure 1.

### 2.2. Dietary Strategies

#### 2.2.1. Time-Restricted Feeding (TRF)

In this study, the 16:8 method of TRF was used. In this dietary strategy, food intake takes place in an 8 h time window and fasting takes place in the remaining 16 h [10]. The subjects were allowed to eat ad libitum during the 8 h meal period and were not given any guidelines on the quantity of food. Within the 16 h fasting period, only calorie-free drinks such as coffee, tea, calorie-free soft drinks, and water may be consumed. In addition, a general nutrient distribution of carbohydrates, fats, and proteins was given according to the recommendations of the International Society of Sports Nutrition and the Institute of Medicine [7,22]. Consequently, approximately 45–65% of the total energy intake should be from carbohydrates, 20–35% from fats, and 20–35% from proteins. The time for food intake was set at 12:00 p.m. till 8:00 p.m. In addition, there was a 30 min buffer time, which subjects were allowed to use at their discretion either before or after the time slot to accommodate any appointments or unknown events. Each food or meal was entered into the food diary with its volume, macronutrient distribution, and energy balance. In addition, the days on which the time window had to be changed or could not be adhered to were also noted.

#### 2.2.2. Macronutrient-Based Diet (MBD)

For the MBD group, all foods were allowed, regardless of the quality of the food, and no time restriction was set. To ensure that the subjects of the MBD group followed a healthy and balanced diet, the 80/20 rule was added to this type of diet. Eighty percent of calories should be consumed via “unprocessed” foods and 20% of calories may be consumed via “processed” foods. As in previous studies, the “Nutri-Score” ranking system was used to categorise foods as “unprocessed” or “processed”, and favourable or unfavourable [23,24,25,26]. Because foods were classified as “unprocessed” only if minimal industrial processing was involved in their preparation (maximum of one or two steps), the macronutrient composition can be considered favourable (less sugar and less saturated fatty acids), and the food ormeal as high in micronutrient content (high in vitamins and/or fibres). On the Nutri-Score scale, these were exclusively foods with an A or B rating. Accordingly, a portion of food is classified as “processed” if more than two industrial processing steps were needed, in which case the macronutrient composition was assessed as unfavourable (no or low content of micronutrients) [23,24,25,26]. The macronutrient ratio was to be rated as unfavourable if the product had a high proportion of animal fats, especially saturated fatty acids (more than one-third percent of fat), and/or had a high content of simple digestible carbohydrates, as well as when energy density concerning the quantity was very high [23,24,25,26]. On the Nutri-Score scale, these foods were ranked exclusively with a D or E. Accordingly, the macronutrient ratio was to be assessed as favourable if the product had a low proportion of animal fats (less than one-third percent of fat) but a high proportion of monounsaturated and polyunsaturated fatty acids, had a low proportion of simple digestible carbohydrates, and the energy density can be classified as low concerning the quantity [23,24,25,26]. Foods with a C on the Nutri-Score scale were rated as neutral and should therefore be avoided.

As MBD is a macronutrient-based approach, the total daily energy expenditure of the subjects must first be determined, for which the Harris-Benedict formula for basal metabolic rate was used, as in recent studies [27].

Male: Basic metabolic rate [kcal/24 h] = 66.47 + (13.7 × body weight [kg]) + (5 × height [cm]) − (6.8 × age [years])(1)

Female:Basic metabolic rate [kcal/24 h] = 65.51 + (9.6 × body weight [kg]) + (1.8 × height [cm]) − (4.7 × age [years])(2)

To approximate the total daily energy intake, we multiplied the calculated value from the Harris-Benedict formula by a physical activity level factor (PAL factor). The PAL factor was derived from the data regarding the subjects’ occupational activity, recreational activities, and sporting activities from the initial interview and ranged between 1.4 and 1.8. As this study involved slightly overweight individuals who had an increased BMI, despite regular physical activity, a calorie deficit of 500 kcal was set during phase 2. The chosen calorie deficit was derived from the recommendations of the International Society of Sports Nutrition [7] and recent studies of Helms et al. [28] and Garthe et al. [29]. In addition, the individual macronutrient requirements for the participants were defined. First, the daily protein requirement was calculated by multiplying body weight by a factor between 1.4 and 2.0 g per kilogram of body weight. For overweight participants, the LBM was used as the basis for calculation and the value of 2.3–3.1 g protein per kilogram LBM was used [7,30]. This factor was selected based on the information on the training frequency of the participants. The higher the weekly training frequency, the higher the chosen multiplier. The factors 1.4–2.0 g per kilogram of body weight and 2.3–3.1 g protein per kilogram LBM were derived from the recommendations of the International Society of Sports Nutrition for building and maintaining muscle mass in athletes [30,31]. After determining the protein requirement of the subjects, fat and carbohydrate requirements were determined by the percentage distribution of carbohydrates 45–65% and fat 20–35% of the total caloric intake. Each food and meal were entered into the food diary with its volume, macronutrient distribution, and energy balance.

### 2.3. Participants

All participants of the study were acquired in a local fitness gym (Windhagen, Germany). They were healthy, physically active, and representative of the general population in Germany, which presents a BMI of 26 and is regularly physically active [32,33]. In addition, subjects had to be between 20 and 40 years old, have a BMI of less than 33, not consume any medication or anabolic steroids, be non-smokers, have no acute injuries or chronic illnesses, and have exercised continuously at least twice a week for 6 months. High-performance athletes with less body fat and individuals trained specifically for national and international competitions were also excluded. 

Before the subjects were included in the study, they were informed about the study design and objectives and had to give their written consent to participate. All personal data collected was anonymised and complies with the data protection regulation in Germany. The entire study design was approved by the local ethics committee of the IST University of Applied Sciences (June 2020) and complies with the Declaration of Helsinki. 

### 2.4. Training

All participants performed at least two training sessions per week at the local gym. The regularity of the training sessions was checked by the investigator through attendance at the gym. The training sessions were divided into three categories before the study: resistance training, class attendance, and endurance training. However, the exact training duration could not be determined and was not considered further. The same frequency and type of training were to be continued throughout the intervention and were checked by the investigator. Additional more or less training sessions were prohibited during the intervention and led to exclusion from the study.

### 2.5. Measurements

#### 2.5.1. Primary Parameters

BW, FM, LBM, BMI, WC, and HC were used as primary measurement parameters. All parameters were collected at the time points T0–T3 in the period 6.00–10.00 a.m. Subjects came fasted and were instructed to drink only 500 mL of water to maintain hydration status. Body composition was measured using an 8-electrode bioelectrical impedance analysis (BIA) from Selvas Healthcare (model: ACCUNIQ BC510, company: Selvas Healthcare Inc., location: HQ 155, Shinseong-ro, Yuseong-gu, Daejeon, 34109 Republic of Korea). Previous studies demonstrated already valid results with this method [34,35,36,37,38].

The waist circumference was measured at the narrowest part of the waist and the hip circumference at the thickest part of the buttocks. The body mass index was determined by the following formula:BMI = body weight (kg)/(body height (m))^2^(3)

#### 2.5.2. Secondary Parameters

In addition to the primary parameters, an adherence diary during the 14 weeks of intervention and a questionnaire were completed at the end of the intervention. The diary was used to check the feasibility of the diet for both groups. The questionnaire was conducted on the influence of the dietary patterns on eating and lifestyle behaviour, for which a 5- or 10-point scale was used. The 5-point scale level offered the following options, for which participants should select the most applicable answer: “fully applicable”, “rather applies”, “partly true”, “hardly applies”, and “does not apply at all”. The 10-point scale level offered an ordinal scale from 1–10, with 1 meaning no consent and 10 representing complete consent. The translated German version of the questionnaire can be found in the Appendix A section.

### 2.6. Statistical Analyses

For statistical analyses, the current version of SPSS (IBM SPSS Statistics 27.0, Ehningen, Germany) was used. All results were presented as mean values with standard deviations (mean ± SD). All parameters were tested for normal distribution at T0 with the Shapiro–Wilk test. Subsequently, all personal data (age, height) and anthropometric parameters (BW, FM, LBM, BMI, WC, HC) were analysed with a 2 × 4 ANOVA (2 groups × 4-time points). In case of group differences, a Bonferroni correction for time × group differences was performed. The training frequencies as well as the data from the diaries and the FDDB Extender App did not show a normal distribution. Consequently, the Mann–Whitney *U* and Kruskal–Wallis tests were used for these parameters. Significant differences were set at *p* < 0.05 and marked with * for time effects and # for group effects. In addition, the effect size (r) was calculated for significant differences using Pearson correlation. Marginal effects were set up to 0.2, small effects up to 0.5, medium effects up to 0.8, and strong effects from 0.8.

## 3. Results

A total of 52 participants were recruited at the local gym. After the initial interview and information about the study design, 42 participants (m = 21, f = 21) took part in the study. Seven subjects (m = 7) were unable to successfully complete the second phase. All three dropouts from the TRF group had to discontinue the study due to illness, whereas the four MBD subjects dropped out for family- and work-related reasons. After the third phase, no further dropouts were recorded, and the data of 35 subjects were analysed. The inclusion and exclusion of the subjects can be seen in Figure 2.

### 3.1. General Data and Training Frequencies

A total of 35 participants were able to successfully complete the study. At the beginning of the study, there were no significant differences between the groups in terms of age, anthropometric data, or training frequency. The participants in the TRF group were 27.9 + 5.3 years old, 173.5 cm tall, and had a BMI of 26.3 kg/m^2^. On average, they exercised 3.7 times/week. In contrast, the MBD group was 27.4 + 5.8 years old, 170.4 cm tall, and had a BMI of 25.7 kg/m^2^. They exercised an average of 4.2 times a week. A detailed list of parameters can be found in Table 1.

### 3.2. Calorie Intake

After analysing the food diaries, we observed no significant differences in total calorie intake and macronutrients between the groups after phase 2. Moreover, no significant differences were observed in total caloric intake of both groups between phase 1 and 2 (phase 1 TRF: 1790.9 ± 440.9 kcal; MBD: 1765.3 ± 377.4) The TRF group consumed an average of 1801.0 ± 421.5 kcal per day during phase 2. Of these, 819.0 ± 185.4 kcal were from carbohydrates, 558.0 ± 172.7 kcal from fats, and 423.0 ± 122.7 kcal from proteins. For the MBD group, an average total kcal intake of 1736.0 ± 419.2 kcal per day was calculated, divided into 818.0 ± 226.6 kcal from carbohydrates, 484.0 ± 118.6 kcal from fats, and 433.0 ± 124.5 kcal from proteins. The detailed list of absolute and percentage values is provided in Table 2.

### 3.3. Primary Parameters

During the two-week familiarisation phase, no significant differences could be detected in any parameters of both groups. Between the measurement time points T0 and T2, the TRF group significantly reduced their mean BW from 80.0 ± 17.1 kg to 76.2 ± 16.6 kg (∆: −3.8 ± 2.1 kg; percent: −4.75%; *p* < 0.000 *; r = 0.413) and the MBD group reduced their BW significantly on average −4.0 ± 2.3 kg, from 74.9 ± 11.6 kg to 70.9 ± 2.3 kg (percent: −5.37%; *p* < 0.000 *; r = 0.346) (Figure 3A). A significant difference was also observed in BMI and FM in both groups between T0 and T2 (Figure 3B,D). The TRF reduced their FM from 22.8 ± 6.4 kg to 19.4 ± 6.0 kg (∆: −3.4 ± 1.6 kg; percent: −14.99%; *p* < 0.000; r = 0.548) and improved their BMI from 26.3 ± 3.1 kg/m^2^ to 25.0 ± 2.9 (∆: −1.3 ± 0.7 percent: −4.94%; *p* < 0.000 *; r = 0.427). The MBD group reduced their FM from 21.5 ± 6.2 kg to 18.6 ± 6.7 kg (∆: −2.9 ± 1.9 kg; percent: −16.13%; *p* < 0.000 *; r = 0.500) and reduced their BMI from 25.7 ± 3.3 kg/m^2^ to 24.4 ± 3.3 kg/m^2^ (∆: −1.3 ± 0.7; percent: −5.06%; *p* < 0.000 *; r = 0.413). For LBM, however, no significant change could be observed in either group. In both groups, LBM remained consistent during the intervention (Figure 3C). The circumference measurements, WC and HC, also showed a significant reduction from T0 to T2 for both groups. The TRF group reduced their WC from 80.6 ± 11.8 cm to 75.8 ± 11.7 cm (∆: −4.8 ± 1.8 cm; percent: −6.0%; *p* < 0.000 *; r = 0.412) and HC from 101.1 ± 5.9 cm to 96.8 ± 5.6 cm (∆: −4.3 ± 1.7 cm; percent: −4.26%; *p* < 0.000 *; r = 0.756). The MBD group achieved a similar change. They reduced their WC from 78.7 ± 10.2 cm to 73.8 ± 9.8 cm (∆: −4.9 ± 2.2 cm; percent: −6.27%; *p* < 0.000 *; r = 0.478) and HC from 101.1 ± 5.9 cm to 96.8 ± 5.6 cm (∆: −4.3 ± 1.7 cm; percent: −4.22%; *p* < 0.000 *; r = 0.724). All anonymised individual changes of the primary parameters are shown in Appendix A.

Furthermore, a significant difference between T0 and T3 in BW, FM, BMI, WC, and HC was observed after the third phase. Between T2 and T3, however, no change could be detected in any of the primary parameters. The changes of all primary parameters between T0 and T3 are shown in Figure 3.

### 3.4. Secondary Parameters

#### Adherence

The evaluation of the adherence diary showed that the TRF group was able to adhere to the specified mealtime window of 12:00 p.m. and 8.00 p.m. on average 6.9 ± 0.2 days per week within the 8-week intervention phase (55.1 ± 1.1 days of 56 days). This corresponds to an adherence to the time window of 98.4%. In contrast, following the analysis of the adherence diary of the MBD group, only 88.9% adherence was observed. This corresponded to 6.2 ± 0.6 days per week (49.8 ± 4.9 days of 56 days) and was significantly lower than the TRF group (*p* < 0.000; r = 0.631). Evaluation of the independent feasibility of each diet form via self-report questionnaire revealed that the adherence of the TRF group decreased by 27.0% to 71.0%. In total, the TRF was complied with on 29.8 ± 11.9 days of the 42 days. This corresponded to a weekly average of 5.0 ± 2.0 days. In comparison, adherence in the MBD group decreased by 23.9% to 65.0%. On average, the subjects were able to adhere for 27.2 ± 7.7 days, which corresponded to a weekly average of 4.5 ± 1.3 days. A group difference could not be observed in phase 3 (*p* = 0.163). In Figure 4, the adherence of both groups over14 weeks is shown.

### 3.5. Questionnaire

After the 14-week intervention, a questionnaire was used to assess the influence of the FDDB Extender Diary documentation on diet. In total, 61% percent of the TRF group answered that the documentation had a positive influence on nutrition and 39% answered that it had little to no influence. In contrast, 82% of the MBD group reported a positive influence. The exact percentage distribution can be found in Figure 5.

In answer to the question of how the diet could be implemented in everyday life, the TRF group reported an average score of 7.26 out of 10. The MBD group had an average score of 8.4 out of 10. In response to the question of whether the documentation of nutrition stressed them in everyday life, about one-third of the TRF group stated that the documentation stressed them (34%). For 66%, documentation was not an additional stress factor in everyday life. In the MBD group, only 12% of the respondents felt that the documentation was an additional stress factor. The exact distribution of the five scales can be seen in Figure 6. 

To the question of “Would you do the diet again for fat reduction”, 94% (17/18) of the TRF group answered yes and 100% of the MBD group answered the same. When asked how long they would follow the diet, the TRF group’s answers averaged 9.7 ± 3.0 weeks and the MBD group 8.9 ± 4.2 weeks. Finally, all participants were asked if they would recommend their diet to others, 89% (16/18) of the TRF group and 100% of the MBD group answered with yes.

## 4. Discussion

This study aimed to compare MBD with TRF in terms of body composition change and sustainability of each approach in healthy, active individuals. The results of this study clearly show that both diets, TRF 16:8 and MBD, significantly reduce the primary measures of BW, BMI, FM, WC, and HC over the 8-week intervention period and maintained LBM. No group difference was found at any time during the study. However, a difference between the two diets was observed in adherence during the 8-week intervention phase. In addition, it was observed that the reduction of the BW, FM, and BMI, as well as WC and HC, also ended with a decrease in adherence. 

When comparing the effects of the TRF group with previous studies, a similar effect on BW, FM, BMI, WC, and HC was found after 8 weeks [12,13,14,15,16,17]. A decisive factor for the effectiveness of TRF is the adherence and the time-limiting window for food intake. Both the results of the third phase and previous studies showed that a low adherence (≤70.0%) cannot be expected to reduce the primary parameters. However, a plateau in the reduction of parameters can be ruled out based on the relatively high values at T2 and the relatively short intervention phase of 8 weeks (phase 2). It can be assumed that a further reduction of the primary parameters could have occurred with an adherence of over 70.0%. Nevertheless, other factors played a role in adherence and implementation with the TRF. Even though the participants in the group would recommend the diet to others and would implement it for an average of just under 10 weeks, a significant drop in adherence was observed in the independent phase 3 without specific support (Figure 4). Psychological and social aspects of affiliation likely play a role in the feasibility of the dietary strategy [11]. Furthermore, it was determined that, despite the increased physical activity, LBM remained the same on average (Figure 3D). However, when looking at the individual changes, it was noticed that some participants lost significant LBM and others gained LBM (Appendix A). Nevertheless, a strong difference in protein intake per kilogram of body weight between individuals showing an increase and those showing a decrease in LBM could not be detected. Both subgroups of the TRF had a protein intake of 1.9 g protein per kilogram LBM, which corresponded to the recommendations of the International Society of Sports Nutrition for individuals with regular physical activity [30,31]. Consequently, other individual factors influence the maintenance, loss, or increase of LBM. Therefore, a clear statement on the maintenance of LBM in TRF with sufficient protein intake cannot be made. In contrast to TRF, there are limited data on MBD with calorie restriction. Initial studies examined only the macronutrient distribution and total calorie intake, but not the change in body composition [9]. Similar to TRF, individual differences in LBM were also observed in MBD. In this group also, significant losses but also increases in LBM were observed despite calorie restriction (Appendix A). Furthermore, no difference in protein intake per kilogram BW was observed. As with the TRF group, it can be assumed that not every individual has an appropriate diet to reduce FM without losing significant LBM. Similar to previous studies, neither dietary strategy has a clear advantage against the other. Interestingly, it was discovered that, without any caloric intake guidelines, a calorie deficit of 400–500 kcal per day was observed in the TRF group, which is in line with the observations of previous research [12,13,14,15,16,17], as well as the recommendations of the International Society for Sports Nutrition for weight reduction [7]. It can therefore be assumed that a small to moderate weight and fat loss can be achieved in 8 weeks either through a defined calorie restriction or through a time-limited food intake. However, regardless of the dietary strategy, it seems that an adherence should be about 90% to be effective. It can also be assumed that TRF requires a higher adherence to achieve the same effect as MBD and that no further changes in body composition occur at an adherence of 65–70%. In addition, both dietary strategies were primarily regarded as positive by the subjects, and they would repeat and recommend them to others. Because both dietary strategies have a positive effect on body weight and fat mass, and, at the same time, the participants evaluated both dietary strategies as positive, it can be assumed that both dietary forms can also be used in clinical studies.

## 5. Limitations

Besides the clear results of this investigation, it also has a few limitations. Firstly, the determination of the subjects’ energy intake using the Harris-Benedict formula and the PAL factor may deviate from the actual needs. This calculation of energy requirements can only be an approximate determination. The use of spiroergometric could have led to a more accurate calculation but was not feasible in this design with this sample size. Another factor was the recording and documentation of the diet using a food diary, as this could also lead to inaccuracies. Nevertheless, the recording and documentation of the diet were based upon previous studies to achieve the best possible comparability [11,12,14,17,18,19,21]. In addition, blood tests of cholesterol (HDL, LDL), oxidative stress, inflammation parameters (interleukins), or glucose concentration as further independent parameters could have supported the positive effects of the nutritional strategies. Furthermore, due to the length of phase 2 (8 weeks), no statement can be made about how long the positive effects of both dietary strategies last. Moreover, with a period of 8 weeks, it is not possible to declare this as a long-term effect. Consequently, future studies should analyse not only anthropometric parameters, but also biological parameters, and determine the current energy requirement using spiroergometric analysis. Nevertheless, important findings on the effects of and adherence to the two dietary strategies could be established in physically active individuals. However, no statements can be made about other population groups, and therefore future studies should examine the effects of both nutrition strategies in different subgroups, such as competitive athletes, to make further statements. 

## 6. Conclusions

In summary, the results of this study demonstrated a similar small to moderate effect of both diet strategies for BW, FM, BMI, HC, and WC after 8 weeks of intervention. However, adherence to both dietary strategies played an important role. Even though adherence differed significantly in phase 2, similar results were achieved. As adherence decreased below 70%, there seemed to be no change in body composition. Furthermore, both dietary strategies were accepted by the participants, and they would recommend them to others. Therefore, both dietary strategies can be used to effectively reduce BW, FM, BMI, HC, and WC. Nevertheless, further studies are essential to make clearer statements on the use of these dietary strategies for different subgroups.

## Figures and Tables

**Figure 1 nutrients-13-03122-f001:**
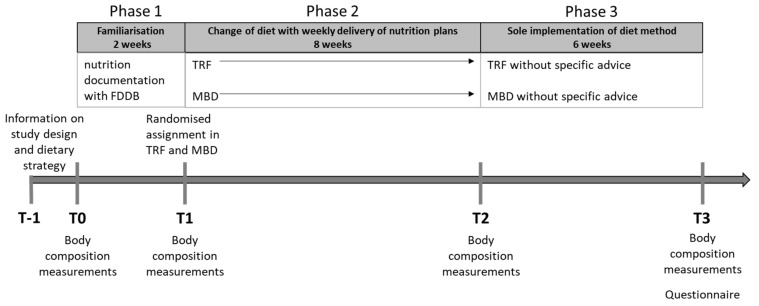
Study design of the intervention with three different phases. Phase 1: Familiarisation with nutrition documentation; phase 2: nutritive investigation (TRF or MBD) with specific support; phase 3: sole implementation of diet method without specific advice. Abbreviation: MBD = macronutrient-based diet; TRF = time-restricted feeding; T = time point.

**Figure 2 nutrients-13-03122-f002:**
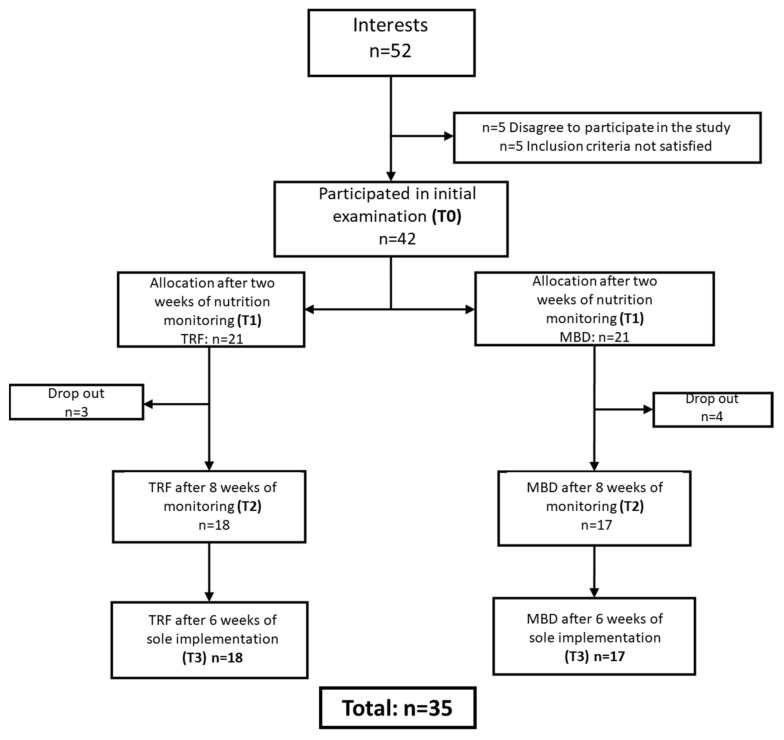
Flow diagram of inclusion and dropout of participants for analysis.

**Figure 3 nutrients-13-03122-f003:**
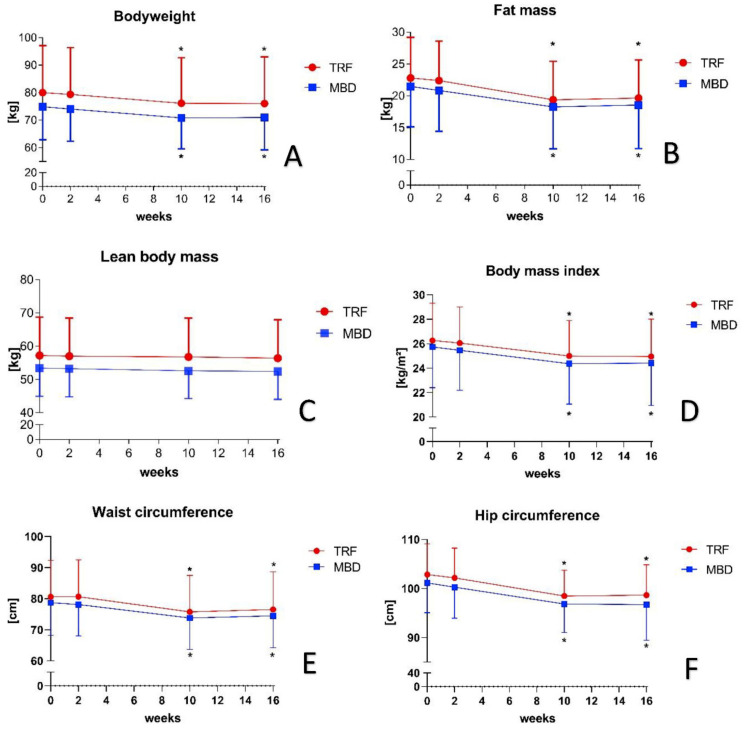
Changes in body composition in the TRF and MBD group ((**A**) body weight; (**B**) fat mass; (**C**) lean body mass; (**D**) body mass index; (**E**) waist circumference; (**F**) hip circumference). Significant differences were set with *p* < 0.05 and marked with * for time differences to T0. Abbreviations: MBD = macronutrient-based diet; TRF = time-restricted feeding.

**Figure 4 nutrients-13-03122-f004:**
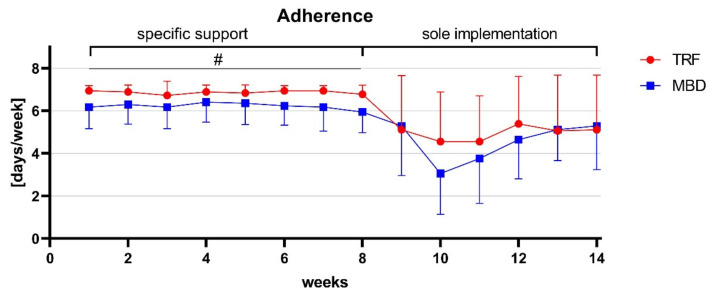
Adherence of the TRF and MBD groups over 14 weeks. Week 1–8: both groups with specific support. Week 9–14: without specific support. Significant group differences were set with *p* < 0.05 and marked with #.

**Figure 5 nutrients-13-03122-f005:**
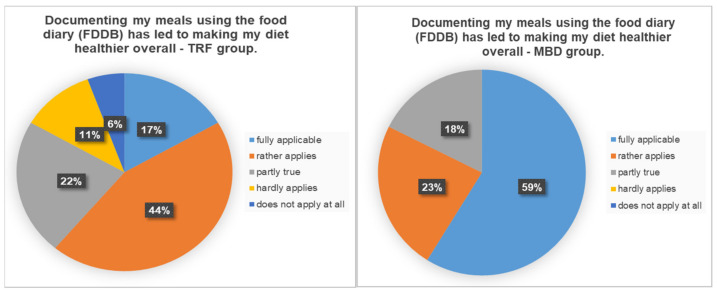
Percentage distribution on the question of whether documenting the meals using the food diary had a positive effect for a healthier diet.

**Figure 6 nutrients-13-03122-f006:**
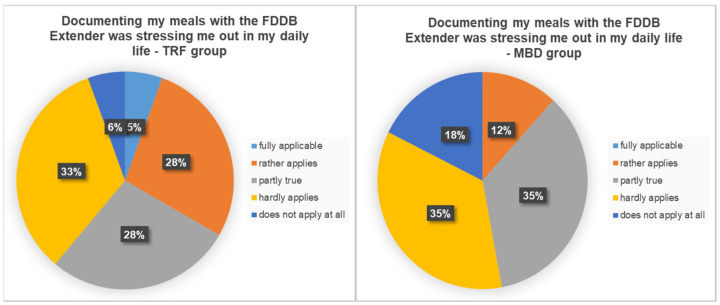
Percentage distribution on the question of whether documenting the meals using the food diary stressed in the daily life.

**Table 1 nutrients-13-03122-t001:** Subject characteristics at time T0 (*n* = 35).

Parameter	TRF (*n* = 18)	MBD (*n* = 17)	*p*-Value
Participants	18	17	
Male	8	6	
Female	10	11	
Age (years)	27.9 ± 5.3	27.4 ± 5.8	0.754
Height (cm)	173.5 ± 10.2	170.4 ± 8.1	0.329
Body weight (kg)	80.0 ± 17.1	74.9 ± 12.0	0.314
Body mass index (kg/m^2^)	26.3 ± 3.0	25.7 ± 3.3	0.628
Fat mass (kg)	22.8 ± 6.4	21.5 ± 6.4	0.549
Fat mass (%)	28.2 ± 3.4	28.5 ± 5.5	0.862
Lean body mass (kg)	57.2 ± 11.5	53.4 ± 8.4	0.272
Waist circumference (cm)	80.6 ± 11.8	78.7 ± 10.5	0.626
Hip circumference (cm)	102.8 ± 6.3	101.1 ± 6.1	0.423
Resistance training (days/week)	2.2 ± 1.0	2.2 ± 1.3	0.766
Endurance training (days/week)	0.6 ± 0.9	1.2 ± 1.6	0.198
Classes (days/week)	1.1 ± 1.2	0.8 ± 1.1	0.520
Total training frequencies (days/week)	3.7 ± 1.5	4.2 ± 1.6	0.386

**Table 2 nutrients-13-03122-t002:** Distribution of macronutrients and total calorie intake of the TRF and MBD group during phase 2.

Macronutrient	TRF	MBD	*p*-Value
Kilocalories (kcal/day)	1801.0 ± 421.5	1736.0 ± 419.2	0.562
Carbohydrate (kcal/day)	819.0 ± 185.4 (45.5 ± 3.4%)	818.0 ± 226.6 (47.1 ± 4.5%)	0.990
Fat (kcal/day)	558.0 ± 172.7 (30.9 ± 4.6%)	484.0 ± 118.6 (27.9 ± 3.9%)	0.154
Protein (kcal/day)	423.0 ± 122.7 (23.5 ± 3.7%)	433.0 ± 124.5 (24.8 ± 2.9%)	0.881

## Data Availability

The data presented in this study are available on request from the corresponding author.

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
