# Peer review of "The Effects of a Macronutrient-Based Diet and Time-Restricted Feeding (16:8) on Body Composition in Physically Active Individuals—A 14-Week Randomised Controlled Trial"

_nutrients, 2021, doi:10.3390/nu13093122_

Round 1

Reviewer 1 Report

Overall, a well written manuscript with a novel area of interest within the space of time-restricted feeding. 

Introduction

Comment 1

Line 57: An explanation as to what the 16:8 method refers to would be beneficial. This is discussed in the discussion section, however a definition in the introduction would be useful.

Methods

Comment 2

Line 86: Who informed the participants about the basics of nutrition? A dietitian/nutritionist? Please clarify. 

Comment 3

Line 96/97: How many days per week were the food diaries completed?

Comment 4

Line 126: In my opinion it would be useful for the reader to clearly state that the MBD had no time restriction on when food could be consumed. 

Comment 5

Line 131: Greater explanation needed are what ‘less industrial processing’ refers to.

Comment 6

Greater description is needed surrounding the definition of what a favourable/unfavourable macronutrient composition is. Furthermore, what does a high/low micronutrient content refer to? What are the cut off points you are using, and are the same for each micronutrient or different?

Comment 7

Line 188: Please clarify the timeframe during which the at least two training sessions occurred, i.e. per week? per fortnight?

Comment 8

Line 233: should state strong effects from 0.8 (currently reads 08).

Results

Comment 9

Line 253 (3.2 Calorie Intake): It is unclear if the results presented in this section is the data following phase 1 or at the end of phase 2. Please clarify.

Comment 10

Line 253 (3.2 Calorie Intake): Whilst discussing total calorie intake and macronutrients, please provide a statement on the overall calorie deficit experienced by both groups during Phase 2 (in comparison to intake during Phase 1).

Comment 11

Line 261 (Table 2): It appears that the title for Table 2 is incorrect, as it currently reads standardized breakfast but is referred to in text as values for overall intake. Also, a bracket is missing in the table when describing the values for Protein.

Comment 12

Line 301: Please provide an explanation on how adherence was measured in the MBD group.

Discussion

Comment 13

In my opinion, the discussion section would be strengthened from a section discussing the clinical significance of the weight loss achieved in both groups.    

Comment 14

Line 386: The statement on adherence needing to be 90% needs further clarification/discussion. What is this based on? Only your research, or other research? Please provide additional information to support this statement, including citations if relevant.

Limitations

Comment 15

It would be worth considering if the length of the intervention (8 weeks) may have impacted any potential changes in primary and secondary outcomes points.

Author Response

Dear Reviewer, 

Thank you for reviewing our manuscript and your important comments. Please find attached the point by point response of both reviewers.

Reviewer 2 Report

Interesting article. Needs careful revision as a large number of details need to be corrected/adjusted. Revise english language.

  1. “The TRF method has already been investigated in a few studies with regard to the change of the primary measures body weight (BW), fat mass (FM), body fat percentage (BF%), body mass index (BMI), LBM, waist circumference (WC), and hip circumference (HC) [11–21]. The most studies examined the effects in mainly sick, overweight, or obese persons [11–17].”

Reviewer (R) 1-I. What are the characteristics of the volunteers enrolled in the 4 missing references in the last sentence (in references 18 to 21)?

R1-II. LBM meaning is missing.

  1. “On average, the participants reduced their BW by -2.6% to -4.0%, FM by -4.0% to -5.9%, BMI by -1.09 kg/m2 to -1.15 kg/m2, and WC by -4.46 cm over the course of 8-16 weeks. Moreover, a daily caloric deficit of -200 kcal to -550 kcal was observed by using the different TRF methods. Also, only few studies investigated the TRF method in healthy, normal weight individuals in terms of body composition. Similar effects were observed in overweight participants for the parameters BW, FM, and LBM [18–21].”

R2-I. Please mention references in each of the sentences above.

R2-II.The last sentence is not clear: a) are the effects upon BW, FM, and LBM observed in overweight participants similar?, or b) are the effects upon BW, FM, and LBM observed in overweight participants similar to the ones observed in healthy, normal weight individuals?

If the first option is true, the corresponding sentence (“Similar effects were observed in overweight participants for the parameters BW, FM, and LBM”) should be third, and not fourth, in the paragraph.

If the second option is true, please adjust the sentence.

  1. “ … as well as the handling of the FDDB Extender food diary …”.

R3. Define FDDB.

  1. “…or has a high sugar content of carbohydrates …” and “… low proportion of sugar in the form of carbohydrates …”

R4. Sugar is a carbohydrate. Authors probably refer to simple, digestible carbohydrates.

  1. “The factors 1.4-2.0g per kilogram of body weight and 2.3-3.1g protein per kg LBM was derived from the recommendations of the International Society of Sports Nutrition for building and maintaining muscle mass in athletes [30, 31].”

R5. “Was” should be replaced by “were”.

  1. “… (model: ACCUNIQ BC510, company: Selvas 202 Healthcare Inc. location: HQ 155, Shinseong-ro, Yuseong-gu, Daejeon, 34109 Republic of 203 Korea).”

R6. Adjust punctuation. Location with capital “L”.

  1. A dot is missing in “…and strong effects from 08.”
  2. Figure 2.

R8. “Rectected”?

  1. Table 2.

R9. Table 2 title and “(kcal/day9” should be corrected.

  1. Figure 4 caption.

R10. Change “D: BMI” to “D: body mass index”.

  1. Adjust formatting style: “... decreased by -27 % to 71 % …” versus “… decreased by 23.9% to 65%.”
  2. “Sixty-one of the TRF group answered that the documentation had a positive influence on nutrition …”.

R12. Sixty-one per cent!

  1. Figures 5, 6 and 7.

R13. Figure 5: include a symbol showing the statistical significant difference between the two groups. Figures 6 and 7: remove 0% from MBD graphs.

  1. Correct the Δ value for BW-TRF and WC-TRC in the results section, primary parameters sub-section.
  2. “The aim of this study was to compare MBD with TRF in terms of body composition change and sustainability of each approach in healthy, active individuals. The results of this study clearly show that both diets, TRF 16:8 and MBD, significantly reduce the primary measures of BW, BMI, FM, TU, and HU over the 8-week intervention period and maintain LBM. No group difference was found at any time during the study. However, a difference between the two diets was observed in adherence during the 8-week intervention phase. In addition, it was observed that the reduction of the BW, FM, BMI, as well as WC and HC, also ended with a decrease in adherence.”

R15. Meaning of TU and HU? Please clarify the last sentence.

  1. Clarify the sentence: ”However, a plateau in the reduction of parameters can be ruled out based on the still relatively high values at time T2 and the relatively short intervention phase of 8 weeks (phase 2).”
  2. “However, when looking at the individual changes, it was noticed that some participants lost significant LBM and others gained LBM (see Supplemental Material).”

R16. Include/add Figure number and letters.

  1. Please carefully check the figure title and the placement of figure caption in all 2-image sets included in Figure 3 (and adjust whenever needed).
  2. “Nevertheless, the recording and documentation of the diet was oriented towards previous studies in order to achieve the best possible comparability [11, 12, 14, 17–19, 21].”

R19. ‘Oriented towards’ or ‘based upon’?

  1. Correct the number of the Figure included in Supplemental Material: why number 3?

Author Response

(The authors gave the same response as above.)
